# Ammonia Emission Characteristics of a Mechanically Ventilated Swine Finishing Facility in Korea

**Gwanggon Jo** [1], **Taehwan Ha** [1], **Yu Na Jang** [1], **Okhwa Hwang** [1], **Siyoung Seo** [1], **Saem Ee Woo** [1], **Sojin Lee** [1], **Dahye Kim** [2] **and Minwoong Jung** [1,*]

[1] Animal Environment Division, Department of Animal Biotechnology and Environment, National Institute of Animal Science (NIAS), Rural Development Administration (RDA), Wanju-Gun 55365, Korea; ggjo@korea.kr (G.J.); thha54@korea.kr (T.H.); jyn0316@korea.kr (Y.N.J.); hoh1027@korea.kr (O.H.); seosi@korea.kr (S.S.); znf12345@korea.kr (S.E.W.); rothdn@korea.kr (S.L.)

[2] Faculty of Biotechnology, College of Applied Life Science, Jeju National University, Jeju 63243, Korea; pioioiq10@gmail.com

[*] Correspondence: mwjung@korea.kr

**Abstract:** In this study, we aimed to determine the ammonia emission characteristics through analysis of ammonia concentration, ventilation rate, temperature, and relative humidity pattern in a mechanically ventilated swine finishing facility in Korea. Three pig rooms with similar environmental conditions were selected for repeated experimentation (Rooms A–C). Ammonia concentrations were measured using a photoacoustic gas monitor, and ventilation volume was estimated by applying the least error statistical model to supplement the missing data after measurement at several operation rates using a wind tunnel-based method. The mean ammonia concentrations were 4.19 ppm, and the ventilation rates were 24.9 $m^3$ $h^{-1}$ $pig^{-1}$. Ammonia emissions were calculated within the range of 0.40–5.01, 0.25–4.16, and 0.37–5.68 g $d^{-1}$ $pig^{-1}$ for Room A, Room B, and Room C, respectively. Ammonia concentration and ventilation rate showed a weak negative correlation (r = −0.13). Ammonia emissions were more markedly affected by ammonia concentration (r = 0.88) than ventilation rate (r = 0.31). This indicates that ammonia concentration reduction can be effective in reducing ammonia emissions. The mean daily ammonia emissions, which increased exponentially over the finishing periods, were calculated as 1.78, 1.57, and 1.70 g $d^{-1}$ $pig^{-1}$ for Room A, Room B, and Room C, respectively (average 1.68 g $d^{-1}$ $pig^{-1}$).

**Keywords:** ammonia; emission; finishing; mechanical ventilation; swine

## 1. Introduction

Ammonia ($NH_3$) is a greenhouse gas released by human activities that indirectly contributes to global warming [1–4]. It is a precursor of ammonium ($NH_4^+$), a major constituent of particulate matter with aerodynamic diameters no larger than 2.5 μm ($PM_{2.5}$), which contributes to air pollution [5,6]. As a highly odorous substance, ammonia raises complaints in residential areas and is toxic to human bodies when inhaled in high concentrations [7]. Agricultural activities are a major source of ammonia [8,9], with emissions from livestock farming accounting for 40–80% of total agricultural emissions [8,10–14].

The Clean Air Policy Support System of the Korean National Institute of Environmental Research monitors and publishes annual reports on ammonia emissions in Korea. Of all livestock farming activities, "manure management" results in the highest ammonia emissions. Ammonia emissions from manure management increased from 138 kt $yr^{-1}$ in 2001 (accounting for 60% of total ammonia emissions) to 217 kt $yr^{-1}$ in 2016 (accounting for 72% of total ammonia emissions) [15,16]. The highest ammonia emission is observed during breeding of cows, pigs, and chickens. In particular, pigs

had the highest emission rate at 97 kt $yr^{-1}$, accounting for 45% of total ammonia emissions from manure management, compared with cows (53 kt $yr^{-1}$) and chickens (54 kt $yr^{-1}$) in 2016. Ammonia emissions from pig manure management account for 32% of total ammonia emissions; thus, pig manure management contributes significantly to ammonia emissions in Korea.

Various methods are used to quantify ammonia emissions from pig farms, including the nitrogen balance method, the micrometeorological method, the model method, and the emission factor method [17]. The emission factor method is an intuitive and commonly used method; it was used to quantify ammonia emissions from Korean pig farms in one study based on data gathered in 2006 [18].

Pig farms vary in the breed of pigs they raise, the ventilation type, and the type of manure storage and thus have different ammonia emission patterns [19]. For these reasons, most countries use environment-specific segmented emission factors for ammonia calculations. However, in Korea, emission factors are determined for only four categories of pigs: "growing (less than two months old)", "first finishing (2–4 months old)", "second finishing (4–6 months old)", and "sow (more than six months old)". In addition, there may be a high probability of uncertainty because the emission factor method was used without considering the ventilation type (i.e., natural vs. mechanical) or method of manure management used. Furthermore, no additional research has been conducted on pig manure management emission factor since 2006.

To improve the ammonia inventory and emission factor, (1) a mechanically ventilated finishing farm was selected; (2) emission patterns were examined by monitoring ammonia concentration, ventilation rate, temperature, and relative humidity at a resolution of one hour throughout the finishing periods; (3) the ammonia emission factor was calculated per pig based on ammonia concentrations and ventilation rates; and (4) previous studies conducted in other countries and results of this study were compared.

## 2. Experiments

### 2.1. Housing

The experiment was conducted in a mechanically ventilated swine farm located in Danyang-gun, Chungcheongbuk-do, Korea, during the finishing periods in the summer and the fall. The cross section and the floor plan of the pig shed in which the experiment was conducted are shown in Figure 1. The shed consisted of eight pens (W555 cm × D208 cm × H260 cm), with four pens placed on either side of the central aisle. Three windows were installed on one side of the sheds for lighting. Outside air entered through the ventilated ceiling above the shed, circulated inside the shed, and exited through a single exhaust fan installed at the center of the ceiling. The floor of the pens comprised a concrete floor and a plastic slatted floor in a 1:1 ratio. An automatic feeder was placed on the plastic slatted floor to prevent the accumulation of feed dropped by the animals over time. A total of four feeders were placed between the pens.

The height of the slurry pit for collecting manure was 45 cm. Pig slurry was collected as soon as the pigs entered the sheds, and it was not flushed during the finishing period. At the end of finishing, the slurry was flushed from both slurry pits simultaneously once the pigs were released. There is a total of 14 fattening pig sheds on the farm. Among them, three sheds with identical structures and environmental conditions (hereafter referred to as "Room A, Room B, and Room C") were selected for repeated experimentation. Rooms A and B face each other on the basis of a narrow corridor, and Room A and C are sheds built with walls interposed therebetween.

### 2.2. Animals

The pigs used in this experiment were 10-week-old finishing pigs, all of which were raised in the sheds for 83 days (Table 1). All pigs were bred under the all-in/all-out production system. There were 91 pigs in Room A, 96 in Room B, and 102 in Room C. The mean weights of the pigs at the point of entry into the rooms were 27.9, 28.3, and 27.1 kg, respectively. The number of male pigs was 24 in all

rooms, and the number of female pigs was 67, 72, and 78 for Rooms A, B, and C, respectively. Ten to twelve finishing pigs were placed in each pen. The mean stocking density was 0.96 m$^2$ pig$^{-1}$.

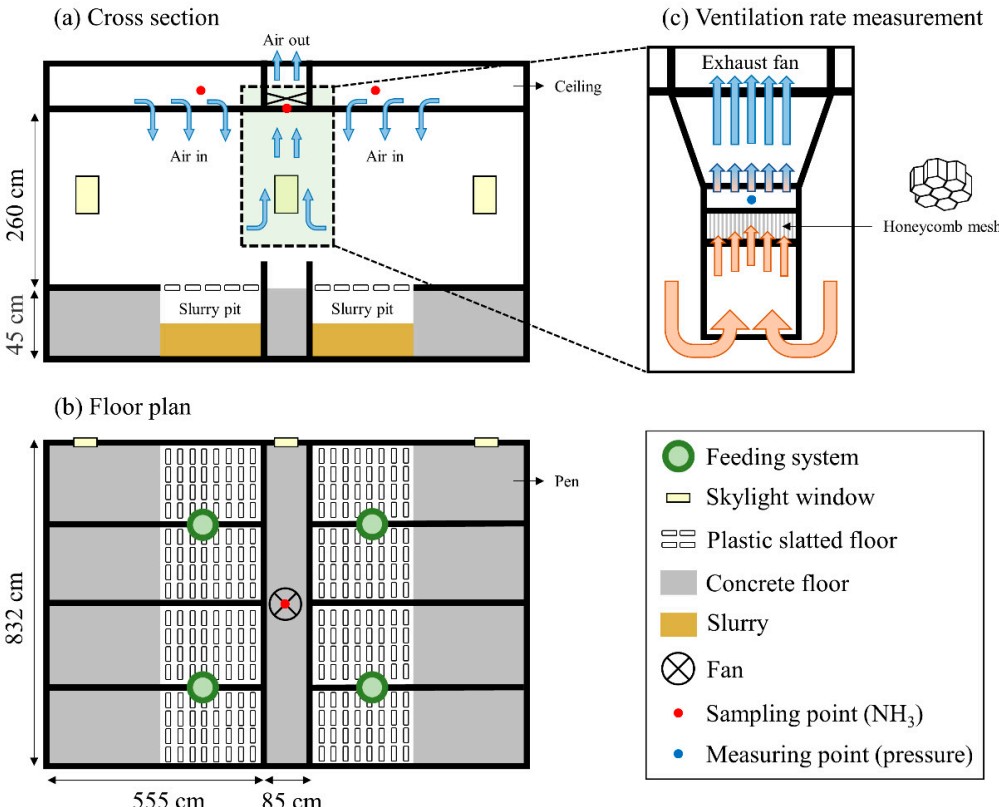

**Figure 1.** (**a**) Cross section, (**b**) floor plan of each room (shed), and (**c**) equipment for ventilation rate measurement used in this study.

**Table 1.** Information regarding number, finishing periods, weight range, mean feed intake, and daily weight gain of the pigs in Rooms A to C. The average feed conversion efficiency was calculated to be 1.97 kg/kg.

| | No. of Pigs | Finishing Periods(Days; Start/End) | Weight Range (kg) | Mean Feed Intake (kg d$^{-1}$ pig$^{-1}$) | Weight Gain (g d$^{-1}$ Pig$^{-1}$) |
|---|---|---|---|---|---|
| Room A | 91 | 83; 15 August–5 November | 27.9–92.0 | 1.55 | 772 |
| Room B | 96 | 83; 22 August–12 November | 28.3–92.2 | 1.54 | 770 |
| Room C | 102 | 83; 29 August–19 November 19 | 27.1–90.4 | 1.43 | 763 |
| Average | 96 | 83 | 27.8–91.5 | 1.51 | 768 |

The pigs were transferred to the sheds starting with Room A on 15 August 2009 to Room B and Room C at a seven-day interval. The pigs were released from Room A on 5 November 2019, Room B on 12 November 2019, and Room C on 19 November 2019. The mean weights at the time of release were 92.0, 92.2, and 90.4 kg, for the respective sheds (Room A, B, and C). The mean amount of feed intake across the three rooms was 1.51 kg d$^{-1}$ pig$^{-1}$. The mean daily weight gain was 768 g d$^{-1}$ pig$^{-1}$.

The nutritional composition of the feed provided during different phases of the finishing period is presented in Table 2. The feed provided for the first seven weeks after the pigs entered the sheds contained 15.0% crude protein, 3.0% crude lipid, 5.0% crude fiber, 5.0% ash, 0.6% calcium, and 0.5% phosphorus. After seven weeks, the crude lipid content of the feed was increased by 1%, and the calcium content was reduced by 0.1%.

**Table 2.** Nutritional content of the feed used in this study (supplied over two terms).

| Nutritional Content (%) | Finishing Period | |
|---|---|---|
| | 1–35 Days | 36–83 Days |
| Crude protein | 15.0 | 15.0 |
| Crude lipid | 3.0 | 4.0 |
| Crude fiber | 5.0 | 5.0 |
| Ash | 5.0 | 5.0 |
| Calcium | 0.6 | 0.5 |
| Phosphorus | 0.5 | 0.5 |

*2.3. Measurements*

Ammonia was sampled through polytetrafluoroethylene (PTFE) tubes ($3 \times 4$ mm (inside diameter $\times$ outside diameter)) by a multi-point sampler (INNOVA 1409, 6 ports, LumaSense Technologies, Ballerup, Denmark) connected to the center of the ventilation fan in each of the three rooms and two sections of the ventilated ceiling in real time. A photoacoustic gas monitor commonly used in livestock production research (INNOVA 1412i, (UA0976 for ammonia, SB0527 for water vapor), LumaSense Technologies, Denmark) was used to instantly analyze the collected samples [20]. The sample integration time was set to 5 s, and compensation for water vapor interference was considered. A calibration curve was first obtained using standard ammonia gas before installing the equipment in the farm. Ten measurements were obtained per sampling point. The means were taken from the last five measurements, because the gas concentration stabilizes after five injections into the gas monitor [21]. Each measurement took 40–50 s, and the measurement for all sampling points took 38–42 min. Each cycle was set to 60 min so that the monitor stopped after 60 min. This is in accordance with the ammonia emission assessment standard for "livestock housing and management systems" in the "VERA test protocol" [22]. The monitor was validated to test its sensitivity at the end of the experiment. Small differences were found in the measurements obtained before and after the experiment (Figure S1). The minimum detection limit of the monitor was measured at 0.2 ppm. The ammonia concentrations were converted to concentration values at 25 °C and 1 atm to examine the concentration patterns and determine ammonia emissions.

The air inside the sheds was ventilated using a fan (Multifan 4E50, Vostermans ventilation BV, Venlo, The Netherlands), measuring 630 mm in diameter, installed at the center of the ceiling. The operation rate (input voltage/rated voltage; η) was automatically adjusted according to the temperature and collected at a resolution of 0.01 (minimum 0.3, maximum 1.0) once per min. As the fan does not automatically measure the ventilation volume, a small ventilation measurement device using a wind tunnel designed to meet the standards of the "American Society of Heating, Refrigerating and Air-conditioning Engineers" was used to measure the operation rate and the actual ventilation volume [23] (Figure 1c). As the farm's policy does not allow the fan to be run at an operation rate less than 0.3, and because lowering the ventilation volume during the finishing periods could cause respiratory diseases in the animals, measurements were taken with the operation rates set at 0.3, 0.5, 0.7, and 1.0. To minimize airflow disturbances, ventilation volume measurement was performed before pig breeding.

A honeycomb-type mesh was placed in the center of the device to rectify turbulence flow. A micro manometer (DP-Calc 5825, TSI, Shoreview, MN, USA) was used to measure the difference between total pressure ($P_t$, Pa) and static pressure ($P_s$, Pa) and to calculate the dynamic pressure of air ($P_d$, Pa). The dynamic pressure was divided by air density ($\rho$: 1.2 kg m$^{-3}$) to determine the velocity (m s$^{-1}$) of the flow (Equations (1) and (2)). The velocity was multiplied by the surface area of the fan (A, m$^2$) to obtain the hourly ventilation volume (m$^3$ h$^{-1}$; Equation (3)). The mean of the five measurements taken at each operation rate was used in the analysis. The ventilation volumes per hour of Rooms A, B, and C were measured at 1084–4485, 913–4136, and 960–4429 m$^3$ h$^{-1}$, respectively.

$$P_d = P_t - P_s \qquad (1)$$

$$Velocity\ \left(m\ s^{-1}\right) = \sqrt{\frac{2 \times P_d}{\rho}} \tag{2}$$

$$Ventilratoin\ volume\ \left(m^3 h^{-1}\right) = A \times Velocity \times 3600 \tag{3}$$

To convert the operation rate to a ventilation volume at a resolution of 0.01 using the measurements obtained at the four operation rates, a regression model was used to estimate the missing values. According to the information provided by the manufacturer, the ventilation fan exhibits a nonlinear relationship between pressure and ventilation volume. Polynomial regression is generally used when the relationship between independent and dependent variables is nonlinear. However, as the fan's performance may have been degraded over time owing to factors such as dust accumulation and corrosion [24,25], various statistical models were used to test the difference between predicted and actual values. Therefore, one linear (degree 1) and three polynomial (degree 3, generalized additive model (GAM), logistic curve) regression models were selected and compared. The root mean square error (RMSE) and the mean absolute percentage error (MAPE) were used to evaluate the models' accuracy in predicting correct values.

The degree 1 model had the highest errors followed by the GAM and the logistic curve. The degree 3 model predicted the actual ventilation volume with 100% accuracy (Figure 2, Table 3). However, overfitting was observed at high operation rate intervals for Room A ($\eta$: 0.89–0.99) and Room B ($\eta$: 0.94–0.99), resulting in the prediction of ventilation volumes higher than an operation rate of 1.0. The ventilation volumes at high voltage must theoretically be higher than those at low voltage. The degree 3 model was removed because its results contradicted this theoretical relationship between ventilation volume and voltages. Following the review of model performance, the logistic curve model was selected for predicting ventilation volumes.

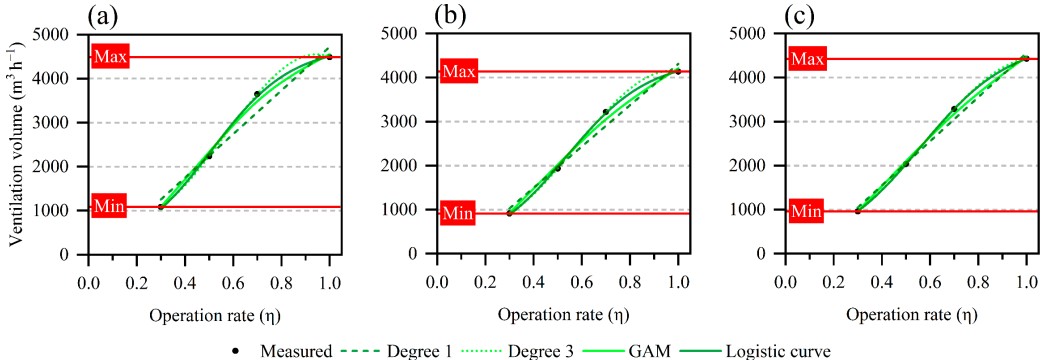

**Figure 2.** Comparison of regression model (degree 1, degree 3, generalized additive model (GAM), and logistic curve) results for ventilation volume according to operation rate ($\eta$) in place in (**a**) Room A, (**b**) Room B, and (**c**) Room C.

**Table 3.** Comparison of RMSE and MAPE of the four regression models (degree 1, degree 3, GAM, and logistic curve) for Rooms A to C.

| Regression Model | Room A | | Room B | | Room C | |
|---|---|---|---|---|---|---|
| | RMSE [a] | MAPE [b] | RMSE | MAPE | RMSE | MAPE |
| Degree 1 | 252 | 8.2% | 190 | 7.0% | 139 | 4.9% |
| Degree 3 | 0 | 0.0% | 0 | 0.0% | 0 | 0.0% |
| GAM [c] | 96 | 2.3% | 101 | 3.1% | 65 | 1.9% |
| Logistic curve | 47 | 2.1% | 28 | 1.5% | 3 | 0.2% |

[a] RMSE: root mean square error. [b] MAPE: mean absolute percentage error. [c] GAM: generalized additive model.

Operation rate and room temperature data were collected once every minute. The thermometer was placed near the shaded center in the shed. The mean operation rate and the room temperature per hour were determined for comparison with ammonia concentration. As a hygrometer could not be placed within the sheds owing to issues such as corrosion and clogging, a moisture filter attached to the photoacoustic gas monitor was used to analyze the dew point of the samples. The Magnus formula was used to indirectly calculate relative humidity, and the mean relative humidity per hour was determined [26]. The coefficients of $\beta$ : 17.62 and $\lambda$ : 243.12 °C ($-45$ °C < temperature < 60 °C) proposed by Sonntag (1990) [27] were used in the calculation.

### 2.4. Data Analysis

The ammonia concentration used in pattern monitoring is the net ammonia concentration, which was determined by subtracting the inlet ammonia concentration ($Ammonia_{in}$, ppm) from the outlet ammonia concentration ($Ammonia_{out}$, ppm). The ventilation rate (m$^3$ h$^{-1}$ pig$^{-1}$) was calculated by dividing the ventilation volume per hour by the number of pigs in each shed (n; Equation (4)). The net ammonia concentration was converted to mg m$^{-3}$ and multiplied by the ventilation rate to determine the measure of ammonia emission per pig per hour ($E_{NH_3}$, g h$^{-1}$ pig$^{-1}$; Equation (5)). The final ammonia emission factor ($EF_{NH_3}$, g d$^{-1}$ pig$^{-1}$) was calculated by dividing the sum of $E_{NH_3}$ measured throughout the finishing period by the number of days in the finishing (d; Equation (6)).

$$Ventilation\ rate\left(m^3\ h^{-1}\ pig^{-1}\right) = \frac{Ventilatoin\ volume}{n} \tag{4}$$

$$\begin{aligned} E_{NH_3}\left(g\ h^{-1}\ pig^{-1}\right) \\ = (Ammonia_{out} \\ -Ammonia_{in}) \times \tfrac{17.03}{24.45} \times V_{flow} \times 10^{-3} \end{aligned} \tag{5}$$

$$EF_{NH_3}\left(g\ d^{-1}\ pig^{-1}\right) = \frac{1}{d}\sum\left\{E_{NH_3}(start\ date) + \cdots + E_{NH_3}(end\ date)\right\} \tag{6}$$

R (version 3.6.3) was used for statistical analysis. The Shapiro–Wilk test and the Anderson–Darling test (for datasets with over 5000 data points) were used to test for normality (*p*-value > 0.05: normal distribution) [28–30]. As the datasets used in this experiment were assumed to have non-normal distributions, Spearman's Rho was calculated for correlation analysis. Outliers were removed according to the VERA test protocol (*Outlier_{upper}* > *upperquartile*(75%) + 3 × *IQR*, *Outlier_{lower}* < *lowerquartile*(25%) − 3 × *IQR*; IQR: inter quartile range). Next, the Kruskal–Wallis test (*p*-value > 0.05: no difference in mean between the groups), which can be used for non-normally distributed datasets, was used to examine differences between the pig sheds [31].

## 3. Results and Discussion

### 3.1. Patterns of Ammonia Concentration, Ventilation Rate, Temperature, and Relative Humidity

Figure 3 illustrates a time–series graph that shows mean daily ammonia concentration, ventilation rate, temperature, and relative humidity. Table 4 shows the mean ± standard deviations of these variables. As the finishing progressed, ammonia concentrations increased, and temperature, relative humidity, and ventilation rates decreased. The mean daily ammonia concentrations were 3.60 ± 2.35 ppm for Room A, 4.23 ± 2.61 ppm for Room B, and 4.73 ± 3.52 ppm for Room C. The daily mean ammonia concentration of the three rooms was 4.19 ± 2.91 ppm. Similar concentrations were reported by Blunden et al. (2008) [32] and Heber et al. (2000) [33] for the summer season (2.45 ± 1.14 ppm and 4.8–6.7 ppm, respectively). Zong et al. (2015) [34], who conducted their study during a similar period to this study (i.e., from August to October), also reported similar ammonia concentrations of 2.1–3.4 ppm. In the present study, the inlet ammonia concentration was 1.62 ppm in

Room A, 1.61 ppm in Room B, and 1.61 ppm in Room C. No significant difference was found in the mean inlet ammonia concentrations of the three rooms.

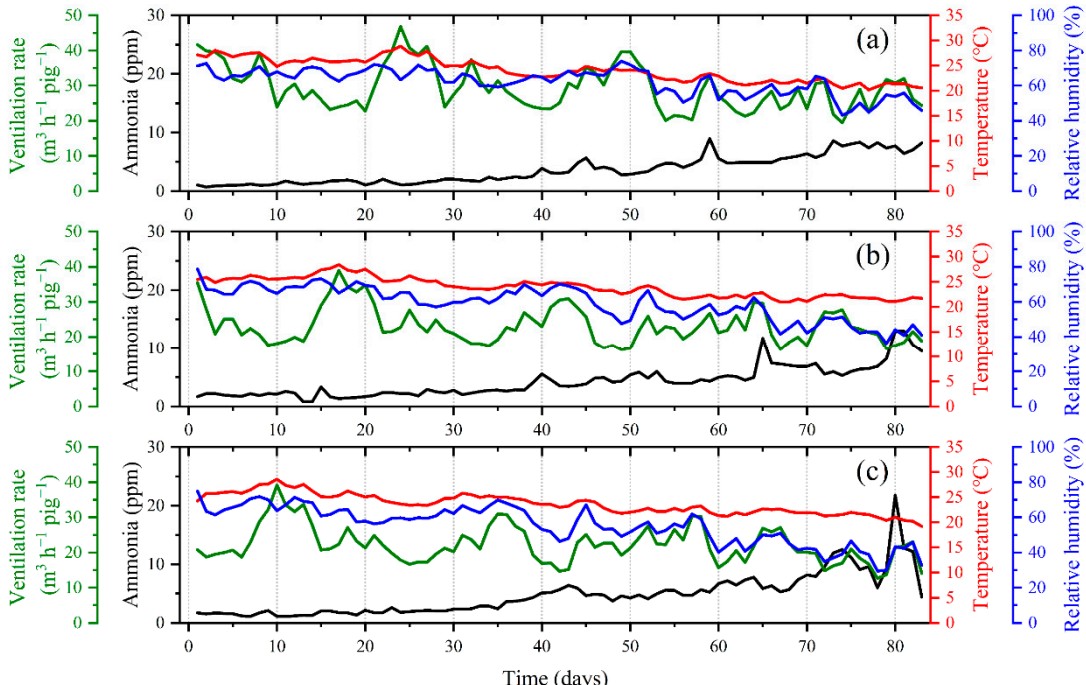

**Figure 3.** Time–series graphs of ammonia concentration, ventilation rate, temperature, and relative humidity in (**a**) Room A, (**b**) Room B, and (**c**) Room C.

**Table 4.** Summary of ammonia concentration, ventilation rate, temperature, and relative humidity (mean ± standard deviation) in Rooms A–C.

|  | Ammonia Concentration (ppm) | Ventilation Rate (m³ h⁻¹ pig⁻¹) | Temperature (°C) | Relative Humidity (%) |
|---|---|---|---|---|
| Room A | 3.60 ± 2.35 | 29.4 ± 6.1 | 24.1 ± 2.3 | 62.1 ± 7.3 |
| Room B | 4.23 ± 2.61 | 23.2 ± 5.0 | 24.0 ± 1.9 | 58.9 ± 9.6 |
| Room C | 4.73 ± 3.52 | 22.2 ± 5.1 | 23.7 ± 2.0 | 55.2 ± 10.7 |
| Average | 4.19 ± 2.91 | 24.9 ± 6.3 | 23.9 ± 2.1 | 58.8 ± 9.7 |

The ventilation rates were $29.4 \pm 6.1$, $23.2 \pm 5.0$, and $22.2 \pm 5.1$ m³ h⁻¹ pig⁻¹ for Rooms A, B, and C, respectively. Thus, Room A had the highest ventilation rate, and Room C had the lowest. The high ventilation rate of Room A may have been because of the fact that Room A pigs were transferred during the high-temperature period of mid-August. The maximum and the minimum ventilation rates throughout the experimental period were observed in Room A (46.8 m³ h⁻¹ pig⁻¹) and Room C (12.7 m³ h⁻¹ pig⁻¹), respectively. The mean temperatures of Rooms A to C were $24.1 \pm 2.3$, $24.0 \pm 1.9$, and $23.7 \pm 2.0$ °C, respectively. The minimum and the maximum temperatures measured during the experimental period were 19.2 °C and 28.9 °C. The temperature decreased by 0.05–0.08 °C per day. The relative humidities of Rooms A to C were $62.1 \pm 7.3$, $58.9 \pm 9.6$, and $55.2 \pm 10.7$%, respectively. The minimum and the maximum relative humidity values were 29.5 and 78.7%. James et al. (2012) [35] reported minimum, maximum, and mean relative humidities of 29, 93, and 67%, respectively, inside pig sheds during the summer.

Hourly data were combined to produce diurnal graphs of changes in ammonia concentrations, room temperature, ventilation, and relative humidity (Figure 4). Data collected over 1992 h were used. Figure 4 shows the graphs for ammonia inlet concentration, ammonia outlet concentration, and ammonia net concentration in the top row and graphs for temperature, ventilation rate, and

relative humidity in the bottom row. A small change in ammonia inlet concentration over time was observed. Ammonia net concentration was most significantly affected by the ammonia outlet concentration, which clearly changed over the course of the day in all the three rooms. It gradually increased from 06:00 to 07:00 h, reached its peak around 13:00 to 14:00 h, and then gradually decreased. Room C had the highest peak concentration, followed by Room B and Room A. Similar patterns were observed for temperature and ventilation rate. Because the ventilation rate changes as the fan's operation rate is automatically adjusted according to the temperature inside the room, temperature and ventilation rate are assumed to be highly correlated. The ventilation rate and the temperature started to increase at 09:00 h, reached a peak at 15:00 h, and decreased thereafter. Temperature showed a smaller range of change over time (22.9–26.2 °C) compared with ventilation rate (17.2–42.7 $m^3$ $h^{-1}$ $pig^{-1}$). Different patterns were observed for relative humidity to those of ammonia concentration, temperature, and ventilation rate. Relative humidity increased at 08:00 h, reached a peak at 13:00 to 14:00 h, and gradually decreased thereafter.

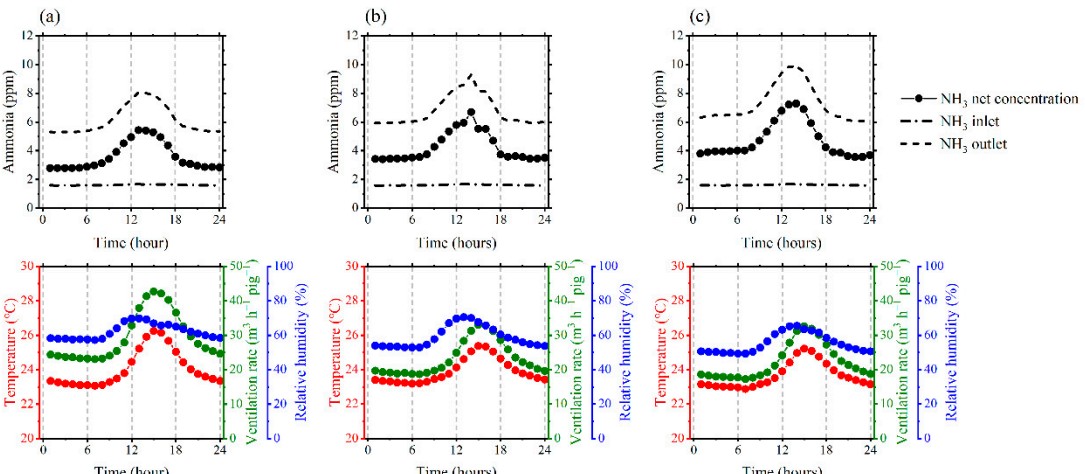

**Figure 4.** Diurnal graphs of ammonia concentration (top), room temperature, ventilation rate, and room relative humidity (bottom) in (**a**) Room A, (**b**) Room B, and (**c**) Room C.

*3.2. Characteristics of Ammonia Emissions*

3.2.1. Time-Series and Diurnal Variation

Daily ammonia emissions per pig, calculated using ammonia concentrations and ventilation rates and the diurnal changes in emissions, are shown in Figure 5. Ammonia emissions exponentially increased as the finishing period progressed. The correlation coefficients between ammonia emissions and the finishing period for Rooms A to C were r = 0.88, r = 0.90, and r = 0.84, respectively. Ammonia emissions and the length of the finishing period were highly correlated with one another, with correlation coefficients greater than 0.8 (Equations (7)–(9)).

$$y = 0.69e^{0.02x} - 0.15 \ (r = 0.88) \tag{7}$$

$$y = 0.23e^{0.03x} + 0.51 \ (r = 0.90) \tag{8}$$

$$y = 1.23e^{0.02x} - 0.82 \ (r = 0.84) \tag{9}$$

Ammonia emissions drastically increased after the feed was changed. The mean correlation coefficient between ammonia emissions and the finishing period was r = 0.82 for the sum of the three rooms. The magnitude of change in the daily ammonia emission decreased to 0.04 g $d^{-1}$ $pig^{-1}$, and the range of change decreased to 0.25–1.74 g $d^{-1}$ $pig^{-1}$ between the first and the 35th day. On the other hand, ammonia emissions highly correlated with the finishing period from day 36 to day 83, and the

magnitude of change in the daily ammonia emission and the range of change increased to 0.18 g d$^{-1}$ pig$^{-1}$ and 0.95–8.38 g d$^{-1}$ pig$^{-1}$, respectively (Figure S2). The changes in the emissions following feed change were attributed to increased food consumption [36].

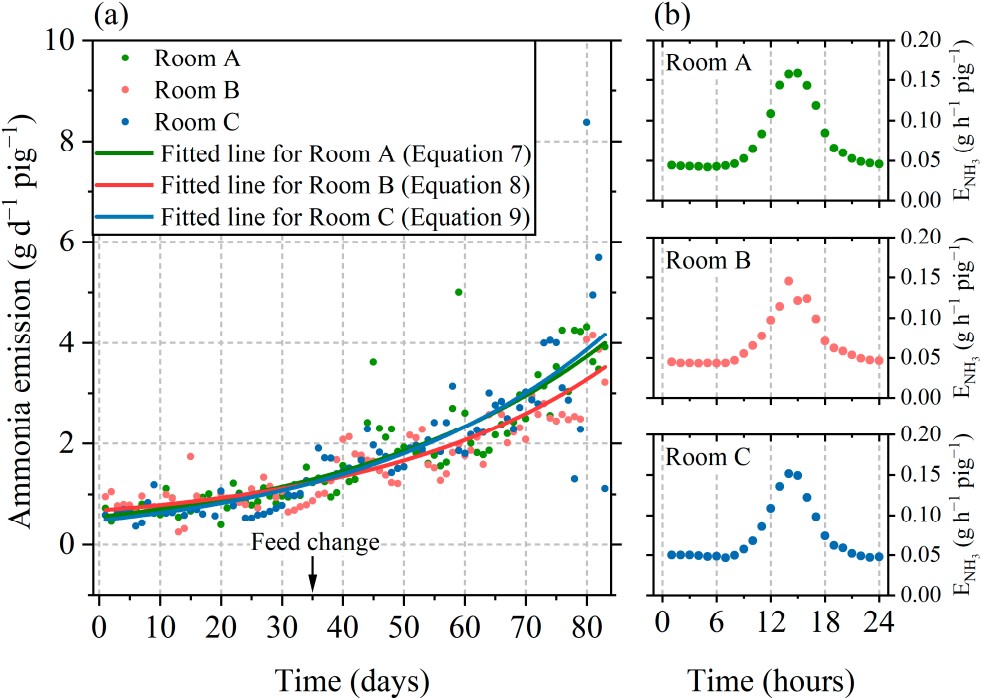

**Figure 5.** (**a**) Time–series graphs of daily ammonia emission in Rooms A to C and their exponential fitted lines and (**b**) diurnal graph of hourly ammonia emissions in Rooms A to C.

Similar patterns in diurnal ammonia emissions were observed across the three rooms (Figure 5b). The diurnal ammonia emissions were 0.04–0.16, 0.04–0.15, and 0.05–0.15 g d$^{-1}$ pig$^{-1}$ for Room A, Room B, and Room C, respectively. Ammonia emissions were generally the lowest at 04:00–05:00 h and peaked at 14:00–15:00 h. Several previous studies reported that pigs are more active during the daytime. The increased activity among the pigs during the day may have affected ammonia emissions [36–40].

3.2.2. Correlation Analysis

Correlations between hourly ammonia concentration, temperature, relative humidity, ventilation rate, and ammonia emission were analyzed (Table 5). Ammonia concentration increased as temperature decreased (r = −0.66). Ammonia concentration was highly correlated with ammonia emissions (r = 0.88) and weakly negatively correlated with ventilation rate (r = −0.13). This finding is consistent with the results of Wi et al. (2019) [41], who found that ammonia concentration within a pig shed was not correlated with ventilation rate. Relative humidity was positively correlated with temperature (r = 0.71) and ventilation rate (r = 0.62). Ventilation rate was positively correlated with temperature (r = 0.58) and relative humidity and weakly correlated with ammonia emission (r = 0.31). In addition, hourly data were combined to produce daily mean data, and the correlation analysis was carried out. Positive or negative correlations were further strengthened, except for the correlations for ventilation rate vs. temperature, ventilation rate vs. relative humidity, and ventilation rate vs. ammonia emission. These stronger negative correlations may be the result of bias reduction due to increased temporal resolution (Table S1) [42,43].

**Table 5.** Results of correlation analyses (Spearman's rho) of hourly ammonia concentration, temperature, relative humidity, ventilation rate, and ammonia emission.

| | Ammonia Concentration | Temperature | Relative Humidity | Ventilation Rate |
|---|---|---|---|---|
| Temperature | −0.66 [a] <br> <0.001 [b] | | | |
| Relative humidity | −0.41 <br> <0.001 | 0.71 <br> <0.001 | | |
| Ventilation rate | −0.13 <br> <0.001 | 0.58 <br> <0.001 | 0.62 <br> <0.001 | |
| Ammonia emission | 0.88 <br> <0.001 | −0.37 <br> <0.001 | −0.12 <br> <0.001 | 0.31 <br> <0.001 |

[a] correlation coefficient. [b] *p*-value.

Figure 6 shows the relationships between ammonia emission and ammonia concentration and ammonia emission and ventilation rate, both of which were highly correlated. Ammonia emissions and its lower limit increased as ammonia concentrations increased (Figure 6a). Low ammonia emissions could not be calculated at high ammonia concentrations. In contrast, there was no clear correlation between ammonia emission and ventilation rate (Figure 6b). Although the range of ammonia emissions was limited at lower ventilation rates, high ammonia emissions were not necessarily observed at high ventilation rates and were observed at low ventilation rates as well. To further analyze the relationships between ammonia concentration, ventilation rate, and ammonia emission, a three-dimensional scatter plot was generated using data from all three rooms, and orthogonal polynomial regression was used to model a surface upon which to map the measurement values (Figure 6c). As the surface is based on two arithmetically determined variables, ammonia concentration and ventilation rate, it can be used for all data using the same equations as those used to determine these two variables.

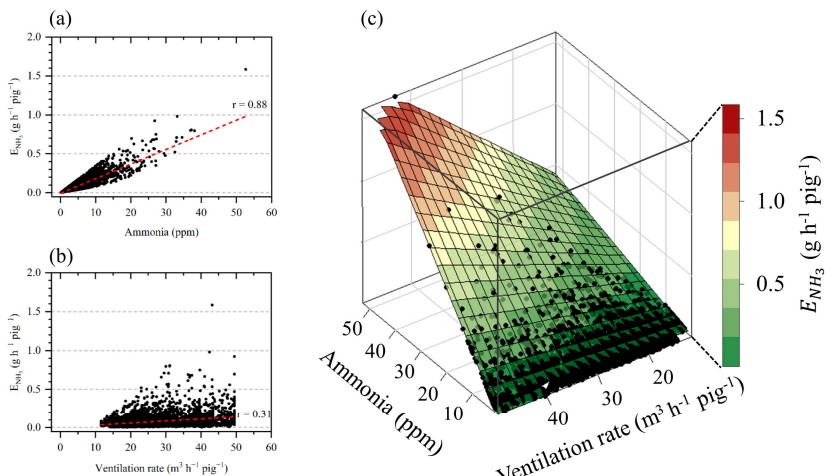

**Figure 6.** Scatter plots of hourly ammonia emission according to (**a**) ammonia concentration and (**b**) ventilation rate. (**c**) Three-dimensional scatter plot of hourly ammonia concentration, ventilation rate, and ammonia emission. The surface plane represents the regression model calculated using Equation (5), and the black circles represent measured data.

The modeled surface represents a space for predicting data and, hence, the interpretation of experimental results is limited to data points on this surface. Data points were highly concentrated across ammonia concentrations of 0–20 ppm and across all ventilation rates. High ammonia concentrations were rarely observed at high ventilation rates, but high ammonia concentrations were also not observed at low ventilation rates. This result suggests that emission control by cleaning out the sheds during hot periods and increasing the frequency of slurry removal could effectively reduce ammonia

emissions. To summarize, when ammonia concentrations are highly correlated with the ventilation rate, ammonia concentrations and emissions can be controlled by changing the ventilation operation rate. However, if the correlation is weak, there is a possibility that ammonia concentrations will vary largely at any ventilation rate. In this case, it may be more appropriate to control ammonia emissions by changing the ammonia concentration rather than the ventilation rate. Nevertheless, the results of this study are based on a single experiment and cannot be generalized to all pig facilities. For a reduction in ammonia emissions, appropriate actions must be accompanied by evaluation and verification of environmental factors for each farm using actual measurements.

### 3.3. Emission Factor

Figure 7 shows a box plot and distribution curve of the daily ammonia emissions from the three rooms. Of the 83 days of the finishing period, two days with outliers were removed, and missing data were estimated with values calculated using regression and Equations (7)–(9). For each box plot, 83 datasets were used, and the percentage of points calculated by equations was 2.4%. The ammonia emissions of Rooms A, B, and C were determined as 0.40–5.01 g $d^{-1}$ pig$^{-1}$ (mean 1.78 g $d^{-1}$ pig$^{-1}$), 0.25–4.16 g $d^{-1}$ pig$^{-1}$ (mean 1.57 g $d^{-1}$ pig$^{-1}$), and 0.37–5.68 g $d^{-1}$ pig$^{-1}$ (average 1.70 g $d^{-1}$ pig$^{-1}$), respectively. There was no statistically significant difference in the mean ammonia emissions of the three rooms ($p$-value = 0.740; Figure 7a). The mean emission factor of the rooms was calculated at 1.68 g $d^{-1}$ pig$^{-1}$. Ammonia emissions of 0.5–1.0 g $d^{-1}$ pig$^{-1}$ were the most common for all three rooms. Rooms A, B, and C accounted for 31.3, 32.5, and 33.7% of all emissions, respectively. Emissions lower than 2.5 g $d^{-1}$ pig$^{-1}$ were dominant, accounting for 80.9% of total emissions (Figure 7b).

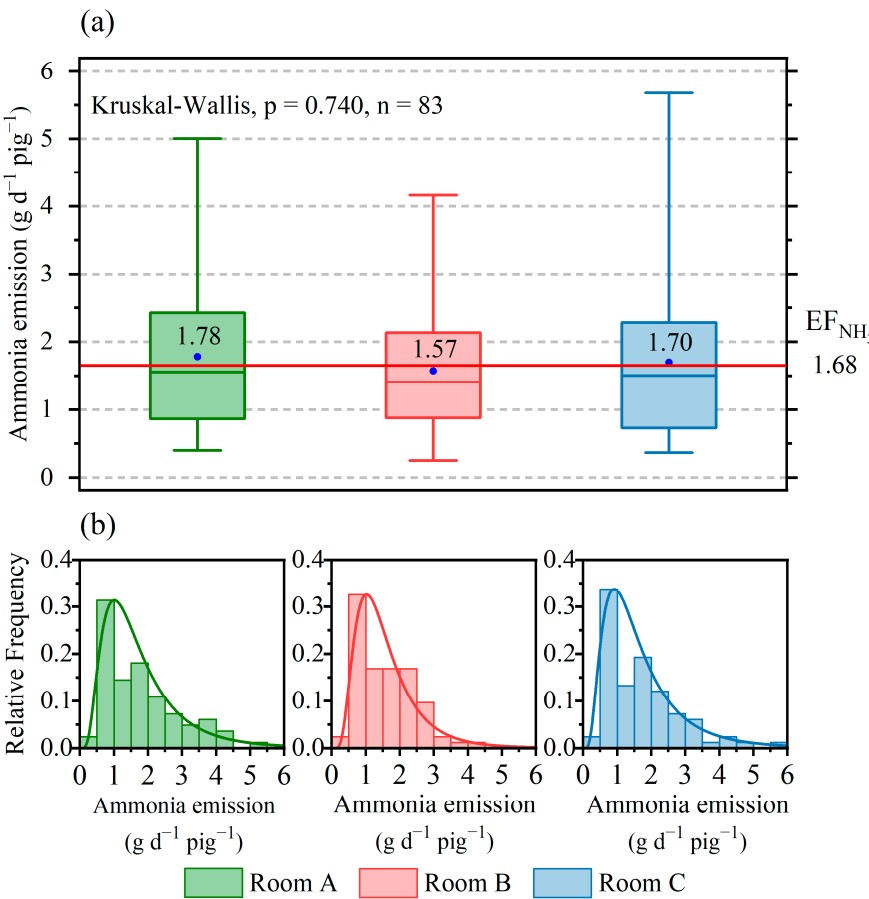

**Figure 7.** (**a**) Box plots and (**b**) distribution curves of daily ammonia emission in each of the three rooms (n = 83). The boxes in (**a**) represent the 25–75% range of samples, and the whiskers show the range

within 3 interquartile range. The horizontal line represents median. Each blue dot is the mean of daily emissions in each room. As a result, the Kruskal–Wallis test showed no significant difference between the mean ammonia emission of the three rooms (*p*-value = 0.740).

Previous studies on ammonia emissions from finishing pigs conducted under similar conditions to this study were reviewed. Table 6 summarizes growing length, number of pigs, weight, ammonia concentration, room temperature, ventilation rate, emission factor, and floor type of this study compared with those of six other studies. Figure 8 shows scatter graphs comparing the relationships between the emission factor and the ammonia concentration and the emission factor and the ventilation rate reported in previous studies. Studies using mass per unit time per 500 kg live mass or animal unit (e.g., g d$^{-1}$ AU$^{-1}$) or mass per time per animal place or space (e.g., g d$^{-1}$ pp$^{-1}$) as the unit of emission factor were excluded, as these units made direct comparison with this study difficult [17]. In other words, the studies that used the same unit as that used in this study were compared. However, 14.1 g d$^{-1}$ AU$^{-1}$ and 1.75 g d$^{-1}$ pp$^{-1}$ could be calculated based on the results (average weight and stocking density) were derived to facilitate comparison with inventories using other units.

**Table 6.** Comparison of pig farm characteristics and results of this study with that of other published studies. Growing length, number, and weight range of pigs, and ammonia concentration, temperature, ventilation rate, ammonia emission factor, and flooring type of finishing swine farms using a mechanical ventilation system are compared.

| Reference | Growing Length | No. of Pigs | Weight (kg) | Ammonia [a] (ppm) | Temperature (°C) | Ventilation Rate [b] (m³ h⁻¹ pig⁻¹) | Emission Factor (g d⁻¹ pig⁻¹) | Floor Type [c] |
|---|---|---|---|---|---|---|---|---|
| [36] [d] | 104 days | 36 | 25.0–111.1 | 7.22 | 23.0 | 53.5 | 5.87 | PS (25%) |
| [44] [e] | 82 days | 25 | 88 | 13.2 | 25.0 | 124.6 | 4.12 | FS |
| [45] [f] | – | 300 | 35– | 15.2 | – | 32.4 | 11.9 | FS |
| [20] | 4 months | 80 | 23.8–111.7 | – | 20.5 | 81.4 | 6.22 | FS |
| [35] [g] | 9 days | 885 | 48.7 | – | 26.0 | 114.1 | 2.94 | – |
| [41] [h] | 14 days | 240 | 80 | 14.9 | 25.0 | 62.0 | 13.8 | FS |
| This study [i] | 83 days | 96 | 27.8–91.5 | 4.19 | 23.9 | 24.9 | 1.68 | PS (50%) |

[a] where the concentration unit was mg m$^{-3}$, it was converted to ppm by applying 24.45/17.03 (assuming 1 atm, 25 °C). [b] where the ventilation rate unit was m$^3$ s$^{-1}$, m$^3$ min$^{-1}$, and m$^3$ d$^{-1}$, it was converted to m$^3$ h$^{-1}$, divided by the number of pigs, and calculated the ventilation rate. [c] abbreviations: PS: partly slatted floor; FS: fully slatted floor. The percentage in brackets is the percentage of slatted flooring. [d] data from summer periods (group 2) were used. [e] the S14 data with the longest experimental period were used. [f] P2(B) data with the smallest number of pigs were used. [g] data from summer periods were used. [h] untreated control data were used. [i] calculated as the average of the three rooms. – not report.

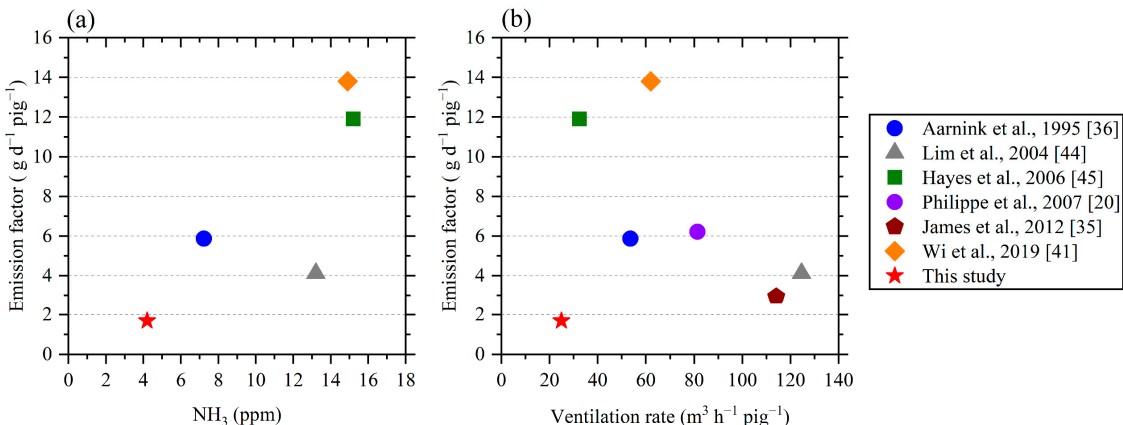

**Figure 8.** Scatter graphs of ammonia concentration, ventilation rate, and ammonia emissions data from previous studies and this study (Table 6). (**a**) Ammonia concentration vs. emission factor and (**b**) ventilation rate vs. emission factor.



Studies varied in length of finishing period (nine days to four months), number of animals bred (25–885 animals), pens that had partly or fully slatted flooring, ammonia concentrations (ranged from 4.19 to 15.2 ppm), room temperature (from 20.5 to 26.0 °C), ventilation rate (from 24.9 to 124.6 $m^3$ $h^{-1}$ $pig^{-1}$), and emission factor (from 1.68 to 13.8 g $d^{-1}$ $pig^{-1}$). This study showed the lowest ammonia concentration, ventilation rate, and emission factor among the compared studies. In general, a higher correlation was observed between ammonia concentration and emission factor. However, James et al. (2012) [35] reported a low emission factor at high ventilation rates, and Hayes et al. (2006) [45] reported a high emission factor at low ventilation rates. Based on these results, ventilation rates may not correlate well with the emission factor. Similar trends were observed when the results of previous studies were comprehensively reviewed without considering the season or the number of animals (Table S2, Figure S3). Nonetheless, as only a few studies were compared, and the studies varied in the number of animals analyzed, these data are insufficient to conclude whether ammonia concentration or ventilation rate increases or decreases ammonia emissions. Further research is needed to analyze the correlations between these factors and ammonia emissions.

## 4. Conclusions

To quantitatively analyze ammonia emissions from pig farming in Korea, a mechanically ventilated swine finishing facility was selected, and experiments were repeatedly conducted for 83 days in three rooms with similar finishing conditions. A ventilation volume measuring device was used to measure the ventilation rate, and a photoacoustic gas monitor was used to analyze ammonia concentrations in real time. Temperature and relative humidity were also monitored in real time. Several statistical models were used to estimate the ventilation volumes at operation rate intervals for which data were missing. An error rate was calculated for each statistical model based on the discrepancy between actual and estimated values. A logistic curve, which most accurately represented actual data, was used to calculate the ventilation rates in this study. A nonlinear statistical model comparison should be considered in cases where the relationship between the operation rate and the ventilation rate is not linear.

The mean ammonia concentrations were 3.60, 4.23, and 4.73 ppm, and the ventilation rates were 29.4, 23.2, and 22.2 $m^3$ $h^{-1}$ $pig^{-1}$ for the respective rooms (Rooms A, B, and C). Ammonia emissions exponentially increased according to the number of finishing days and were measured as 0.40–5.01, 0.25–4.16, and 0.37–5.68 g $d^{-1}$ $pig^{-1}$ for Rooms A, B, and C, respectively. The daily ammonia emission rate was similar across the three rooms. In the correlational analysis, ammonia concentration was negatively correlated with temperature and highly correlated with ammonia emission (r = 0.88). In particular, the minimum limit of ammonia emissions was found to increase consistently with increase in ammonia concentration. On the other hand, ventilation rate weakly correlated with ammonia concentration (r = −0.13) and ammonia emission (r = 0.31). Low ammonia emissions were calculated even at high ventilation intervals. In cases where ammonia concentration has a higher impact on ammonia emission than ventilation rate (given that ammonia concentration was weakly correlated with ventilation rate), ammonia emissions can be effectively reduced by handling the emission source, such as cleaning the pig pens or increasing the frequency of slurry removal.

The mean ammonia emissions were 1.78, 1.57, and 1.70 g $d^{-1}$ $pig^{-1}$ for Rooms A, B, and C, respectively. There was no statistically significant difference between the mean ammonia emission of the three rooms. The emission factor was calculated at 1.68 g $d^{-1}$ $pig^{-1}$ based on average of the three rooms.

**Supplementary Materials:** The following are available online at http://www.mdpi.com/2073-4433/11/10/1088/s1, Figure S1: Results of calibration using photoacoustic spectroscopy equipment, Figure S2: Scatter plots of the total daily average ammonia emissions across the three rooms, Figure S3: Scatter graphs of ammonia concentration, ventilation rate, and ammonia emissions data from previous studies and this study, Table S1: Results of correlation analyses (Spearman's rho) of daily ammonia concentration, temperature, relative humidity, ventilation rate, and ammonia emission, Table S2: Comparison of pig farm characteristics and results of this study with that of other published studies.

**Author Contributions:** Conceptualization, G.J. and M.J.; methodology, G.J., T.H., and M.J.; software, G.J.; validation, G.J., Y.N.J., and S.S.; formal analysis, G.J.; investigation, G.J. and Y.N.J.; data curation, G.J.; writing—original draft preparation, G.J.; writing—review and editing, G.J., T.H., Y.N.J., O.H., S.S., S.E.W., S.L., D.K., and M.J.; visualization, G.J.; supervision, M.J.; project administration, M.J.; funding acquisition, M.J. All authors have read and agreed to the published version of the manuscript.

**Funding:** This research was funded by Cooperative Research Program for Agriculture Science and Technology Development of RDA, project number PJ01385002.

**Conflicts of Interest:** The authors declare no conflict of interest. The funders had no role in the design of the study; in the collection, analyses, or interpretation of data; in the writing of the manuscript, or in the decision to publish the results.

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
