# Peer review of "Ammonia Emission Characteristics of a Mechanically Ventilated Swine Finishing Facility in Korea"

_atmosphere, doi:10.3390/atmos11101088_

Round 1

Reviewer 1 Report

Thank you for your effort, publishing data, and good presentation of data. Here are my comments to improve the paper:

General comments:

  1. Tables and Figures should be placed after it is been discussed in the body of the paper.
  2. Please do not repeat the information/data that is already provided in the Tables/Figures. That seems redundancy and hopefully, save some time of future readers.

Introduction:

  1. line 62: objective 2 is misleading. I think what you mean is correlating ventilation rate with η. Missing data imply that some data were lost and you are trying to recover those while this is no the case. Please consider revising. Needless to say, revise this wording throughout the rest of the paper where you mentioned this again .e.g. line 149.

Methods:

  1. It would be helpful if you mention how many other rooms are there on the farm site. Even better if you could provide a map or relative location of rooms A, B, and C compare to each other and other rooms if there is any.
  2. Figure 1: 
    1. It was initially thought that rooms A, B, and C are one of each pen. So I think there is room for confusion. Please consider more clarification in the title.
    2. Please specify the location of ammonia sampling point on the floor plan.
    3. Please specify the location of fan on the floor plan.
  3. Table 1:
    1. it is not obvious what are those numbers in the parentheses in the second column
  4. Line 97-100: Please specify the year.
  5. Line 138: Please mention when the measurements were done. Was it before placing the pigs on the farm or after or maybe sometime in the middle?
  6. Please mention the location(s) at which temperature was measured as well the instrument.
  7. Line 185: Please double check if n is the number of pigs in each room not pen.

Results:

  1. Figure 4, bottom figures: to better visualize variations within each parameter, please consider adjusting Y-axis range e.g., for temperature a range of 20 to 30 degrees.
  2. Line 240: Please explain what could drive this among rooms.
  3. Line 262: this sentence seems to be vague. It is mentioned that magnitude of change in the daily ammonia emission decreased to X. Decreased from what number?
  4. Section 3.2.2: Correlation analysis: Interpreting the correlation between some of the variables should be discussed with caution. For instance, the correlation of temperature and ammonia concentration could be positive if the study was done in the summer. Clearly, if there is no other factor, this correlation should be positive based on physics principles. As another example, the ventilation rate is correlated with temperature because the fan is set up that way so correlation of ventilation rate and temperature could not add a lot into the context. Lastly, the correlation of relative humidity and temperature is increasing the holding capacity of air which is a known fact. Please reconsider presenting all correlations.
  5. Line 333-334: Please mention how many data point and their percentage estimated using those equations.

Conclusion:

  1. line 386: ventilation volume measuring device development was not presented in the methods. If you mean ventilation rate estimation method, please revise or present that device in the method.

Reviewer 2 Report

This is an excellent study and much-needed data on emission factors for ammonia and swine finishing operations in the Republic of Korea. I have several items for improvement that will make this work more impactful and comparable with emission factors in other countries for the same type of livestock operations:

  • Table 1 caption.  Please add a description to the date formats in the "finishing periods' column.
  • Table 1 - it would be great if the feed conversion efficiency could be added.
  • Table 3 - please add a note with spelled out acronyms.
  • Equation 4 - correct the spelling of 'ventilation'
  • Emission factor units - the 'per h and per pig' emission factor is the obvious one, but much more useful for comparisons with other emission factors for the same type of systems are:
    • mass/day/AU (animal unit defined as 500 kg of live weight). This takes care of the differences in the actual body weight constantly changing. The Authors already have this data - I encourage them to re-analyze it and report it also in this way.  Also, per day is more intuitive.
    • mass/day/animal space (this unit requires the knowledge of the stocking density used).
  • Once the additional EF units are used, the data could be compared with other studies that also report EFs on the basis of AU and animal space. It is worth the effort, as readers will appreciate the thought the Authors put into making the data even more accessible and comparable.
